# The impact of SARS-CoV-2 infection on renal function in patients with biopsy-proven kidney diseases

**Bogdan Obrișcă**[1,2]*, **Valentin Mocanu**[2], **Alexandra Vornicu**[1,2], **Roxana Jurubiță**[1,2], **Bogdan Sorohan**[1,2], **George Dimofte**[2], **Camelia Achim**[1,2], **Andreea Andronesi**[1,2], **Georgia Micu**[2], **Raluca Bobeică**[2], **Nicu Caceaune**[3], **Alexandru Procop**[4], **Vlad Herlea**[1,4], **Mihaela Gherghiceanu**[1,5], **Gener Ismail**[1,2]

1 "Carol Davila" University of Medicine and Pharmacy, Bucharest, Romania, 2 Department of Nephrology, Fundeni Clinical Institute, Bucharest, Romania, 3 Department of Internal Medicine, Fundeni Clinical Institute, Bucharest, Romania, 4 Department of Pathology, Fundeni Clinical Institute, Bucharest, Romania, 5"Victor Babes" National Institute of Pathology, Bucharest, Romania

* obriscabogdan@yahoo.com

## Abstract

### Background

We sought to evaluate the long-term effects of COVID-19 on renal function in patients with biopsy-proven kidney diseases.

### Methods

A total of 451 patients with biopsy-proven kidney disease and at least 12 months of follow-up subsequent to COVID-19 pandemic onset were included in the study. The primary study endpoint was a composite of a persistent decline of more than 30% in eGFR or ESRD.

### Results

23.1% of patients had COVID-19 during a follow-up period of 2.5 y (0.8–2.6), while 17.6% of patients reached the composite endpoint. Those with COVID-19 were more likely to reach the composite endpoint [26.7% vs. 14.8%; OR, 2.1 (95%CI, 1.23–3.58), p = 0.006). There was a significant eGFR change in the first year of follow-up between the two study groups [-2.24 (95%CI,-4.86; 0.37) vs. +2.31 (95%CI, 0.78; 3.85) ml/min, p = 0.004], with an adjusted mean difference of -4.68 ml/min (95%CI,-7.7; -1.59)(p = 0.03). The trend for worse renal outcomes remained consistent in patients with IgAN, MN and FSGS, but not in those with LN. After multivariate adjustment, the independent predictors of the composite endpoint were baseline eGFR (HR, 0.94; 95%CI, 0.92–0.95), COVID-19 (HR, 1.91; 1.16–3.12) and male gender (HR, 1.64; 95%CI, 1.01–2.66). In multivariate linear regression analysis, COVID-19 independently determined a reduction of eGFR at 12 months by 4.62 ml/min/1.73m$^2$ (β coefficient, -4.62; 95%CI, -7.74 to -1.5, p = 0.004).

**Funding:** The author(s) received no specific funding for this work.

**Competing interests:** The authors have declared that no competing interests exist.

## Conclusions

There is a significant impact of COVID-19 on long-term renal function in patients with biopsy-proven kidney diseases, leading to a greater decline of eGFR and a worse renal survival.

## Introduction

The coronavirus disease-2019 (COVID-19) is associated with substantial morbidity and mortality in the acute phase, while certain populations (e.g., elderly patients, those with chronic conditions) appear to be at higher risk for severe forms of COVID-19 [1]. Nonetheless, patients with chronic kidney disease (CKD), including the dialysis and renal transplant recipients' populations, were particularly prone to worse outcomes following severe acute respiratory syndrome coronavirus 2 (SARS-CoV-2) infection from both pulmonary and/or extrapulmonary organ dysfunctions [2–4].

Despite that most studies of patients with CKD focused on mortality as the primary outcome following COVID-19 pneumonia, emerging evidence suggests that SARS-CoV-2 infection is a risk factor for renal events as well, increasing the risk for acute kidney injury (AKI), new-onset or relapsing glomerular disorders and accelerated renal function decline [5–8]. A meta-analysis comprising 30,657 hospitalized patients with COVID-19 identified a pooled prevalence of AKI of 28% (95%CI, 22–34%) and a pooled prevalence of kidney replacement therapy of 9% (95%CI, 7–11%) [9]. Similarly, in a study evaluating over 1.7 million US veterans, of whom approximately 90,000 patients had COVID-19, a 1.94-fold higher risk of AKI, a 1.25-fold higher risk of estimated glomerular filtration rate (eGFR) decline≥30% and a 2.96-fold higher risk of end-stage renal disease (ESRD) were identified on long-term follow-up [7]. In addition, a graded impact of the severity of COVID-19 on the major adverse kidney events was defined [7]. However, in the last study, the eGFR decline or progression to ESRD were evaluated after a median follow-up time of only 164 days [7]. Contrary, a recent report showed that, following non-severe COVID-19, patients had only a slightly lower eGFR, but no signs of progressive kidney injury [10]. While both direct and indirect mechanisms have been proposed to underlie kidney injury in COVID-19, a definitive evidence of the SARS-CoV-2 kidney tropism is lacking [11, 12]. Nonetheless, episodes of AKI are a known risk factor for future CKD, while, in those with preexistent CKD, a SARS-CoV-2 infection might lead to accelerated nephron loss and reduced kidney lifespan [11]. As such, studies with an adequate follow-up period to evaluate the impact of SARS-CoV-2 infection on renal function are lacking.

Accordingly, we sought to evaluate the long-term effects of SARS-CoV-2 infection on renal function in patients with biopsy-proven kidney diseases.

## Material and methods

### Study population and data collection

This is a retrospective, observational study that enrolled all patients that underwent a kidney biopsy in our department between March 2007 and April 2021. The inclusion criteria were: age over 18 years, patients with at least 12 months of follow-up subsequent to COVID-19 pandemic onset (26th February 2020, when the first case of SARS-CoV-2 infection was reported in Romania) and at least 12 months of follow-up following the SARS-CoV-2 infection, patients with at least a yearly assessment of renal function during the follow-up time. The exclusion

criteria were: age under 18 years-old, patients with insufficient clinical data or a follow-up time shorter than 12 months, patients that died or progressed to ESRD prior to COVID-19 pandemic onset, patients without a biopsy-proven kidney disease, patients that did not have a yearly assessment of renal function in our department.

The clinical variables obtained by reviewing the patient's medical records at the time of kidney biopsy were age, gender, Charlson Comorbidity Index, comorbidities (viral infections, arterial hypertension, obesity, malignancies, diabetes, and cardiovascular disorders), therapy with renin-angiotensin-aldosterone system (RAAS) inhibitors and type of immunosuppressive (IS) agents. Laboratory data included renal function assessment by serum creatinine and eGFR (calculated by CKD-EPI equation) at different time-points: study baseline, at 12 months and at the last follow-up visit. eGFR values after initiation of renal replacement therapies were excluded. For patients with a kidney biopsy prior to the COVID-19 pandemic, the baseline renal function was evaluated based on a serum creatinine within 3 months (before or after) of the pandemic onset. Additionally, for those with a kidney biopsy during the COVID-19 pandemic the baseline renal function was evaluated based on the closest serum creatinine to the pandemic onset (if available) or on the serum creatinine from the moment of biopsy.

The study was conducted with the provisions of the Declaration of Helsinki and the protocol was approved by the local ethics committee (The Ethics Council of Fundeni Clinical Institute, Registration number: 8850; date of approval: February 16th, 2021). All patients provided written informed consent before study entry. The access to patient's medical records to collect the data for research purposes was made between 1st January 2022 and 1st September 2022, on a weekly basis (each Friday). The access was needed weekly due to the volume of the data collected and to ensure an adequate follow-up of the patients.

## SARS-CoV-2 infection diagnosis and assessment of severity

The diagnosis of SARS-CoV-2 infection was based on either a positive nasopharyngeal swab, as determined via real-time reverse transcription polymerase chain reaction (RT-PCR) and/or antigen tests, or on a positive serological test. SARS-COV-2 infection screening was undertaken regularly in our center with a periodicity of 1–3 months. Our center's protocol was to undertake a proactive screening for SARS-CoV-2 infection in this category of immunosuppressed patients via outpatient visits, hospital admissions, periodic telephone interviews and a thorough review of the electronic clinical health records during the study follow-up. COVID-19 was defined as participants who reported "definitely" or "probably or suspected" COVID-19 or had a positive antigen or serological test for COVID-19. Patients that never had any suspicion for SARS-CoV-2 infection, had persistent negative testing and negative serologic workup were considered SARS-CoV-2 negative.

SARS-CoV-2 infection severity was assessed as defined by the COVID-19 Treatment Guidelines Panel of the National Institutes of Health as mild illness (individuals who have any of the various signs and symptoms of COVID-19 but who do not have shortness of breath, dyspnea, or abnormal chest imaging), moderate illness (individuals who show evidence of lower respiratory disease during clinical assessment or imaging and who have an oxygen saturation ≥94% on room air at sea level) and severe illness (individuals who have oxygen saturation <94% on room air at sea level, a ratio of arterial partial pressure of oxygen to fraction of inspired oxygen <300 mmHg, respiratory frequency> 30 breaths/min, or lung infiltrates >50%) [13].

Our center's protocol was not to modify the immunosuppressive regimens during the COVID-19 pandemic. However, during a SARS-CoV-2 infection a temporary reduction in the doses of immunosuppressive regimens (for up to 10–14 days following the resolution of the infection) was undertaken.

## Study endpoints

The primary study endpoint was a composite of a persistent decline of more than 30% in eGFR or ESRD (dialysis, renal transplant or eGFR<15 ml/min/1.73m$^2$), whichever came first. The secondary study endpoints were eGFR change at 12 months and eGFR decline per year.

## Statistical analysis

Continuous variables were expressed as either mean (± standard deviation) or median [and either interquartile range, (IQR:25$^{th}$-75$^{th}$ percentiles) or 95% confidence interval (95%CI)], according to the distribution, while categorical variables were expressed as percentages. Differences between groups were assessed in case of continuous variables by Student *t* test, Mann–Whitney test, one-way ANOVA or Kruskal–Wallis test, according to the distribution of dependent variables and the level of independent variable, and in case of categorical variables by Pearson χ2 test or Fisher's exact test.

The probability of event-free survival was assessed by Kaplan-Meier method and the log-rank test was used for comparisons. Univariate and multivariate Cox proportional hazards regression analyses were performed to identify independent predictors of the primary endpoint. The results of Cox analyses are expressed as a hazard ratio (HR) and 95% confidence interval (95% CI). To build the best prediction model, we selected the independent risk factors for the development of the endpoint using a multivariate Cox proportional hazards model with stepwise backward elimination. In addition, univariate and multivariate linear regression analyses were developed to evaluate the impact of SARS-CoV-2 infection on eGFR change at 12 months after pandemic onset. The eGFR change will also be analyzed using analysis of covariance (ANCOVA), with infection occurrence as a fixed effect and the baseline eGFR value as covariate. In all analyses, p values are two-tailed and all p values less than 0.05 were considered statistically significant.

Statistical analyses were performed using the SPSS program (SPSS version 20, Chicago, IL), and GraphPad Prism version 9.3.1 (1992–2021 GraphPad Software, LLC).

## Results

### Study population and SARS-CoV-2 infection characteristics

A total of 749 patients that underwent a kidney biopsy between March 2007 and April 2021 were considered for study inclusion. Of these, 52 patients died prior to COVID-19 pandemic, 87 patients progressed to ESRD before study onset and 159 patients were monitored in other centers or were lost to follow-up, leaving a final cohort of 451 patients for analysis (S1 Fig).

The study population had a mean age of 49.2 ± 14.8 years, while 52.3% of patients were males. At the time of COVID-19 pandemic onset, the study cohort had a mean serum creatinine of 2 ± 1.47 mg/dl and eGFR of 52.1 ± 29.7 ml/min/1.73m$^2$, respectively (Table 1). In terms of comorbidities, 13.5% of patients had chronic viral infections, 16% had diabetes mellitus, while 75.2% had arterial hypertension. Most of the patients (76.7%) received RAAS inhibitors, while 71.8% had received various IS regimens.

The study cohort consisted of mainly glomerular disorders (84%), the most prevalent being IgA nephropathy (IgAN) (22.6%), followed by lupus nephritis (LN) (11.5%) and membranous nephropathy (MN) (9.5%) (S1 Table, S2 Fig). Tubulo-interstitial disorders accounted for 6.9% of cases, while vascular disorders (ischemic and hypertensive nephropathy) accounted for 7.1% of cases.

During a median follow-up period of 2.5 y (IQR: 0.8–2.6), 23.1% (n = 104) of patients had a SARS-CoV-2 infection (Table 1). In patients with SARS-CoV-2 infection, the median post-

**Table 1. Characteristics of study cohort.**

| Variable | Entire cohort | With COVID-19 | Without COVID-19 | p-value |
|---|---|---|---|---|
| Number of pts. | 451 | 104 | 347 | |
| Age at biopsy (y) | 47.5 ± 14.9 | 44.8 ± 12.8 | 48.2 ± 15.4 | 0.04 |
| Age at study onset (y) | 49.2 ± 14.8 | 46.7 ± 12.6 | 50 ± 15.3 | 0.03 |
| Gender (%M/%F) | 52.3%/47.7% | 51%/49% | 52.7%/47.3% | 0.75 |
| Charlson Comorbidity Index | 2 (0–6) | 2 (0–5) | 2 (0–6) | 0.86 |
| SCr at study onset (mg/dl) | 2 ± 1.47 | 2.12 ± 1.49 | 1.97 ± 1.47 | 0.36 |
| SCr at last FU (mg/dl) | 2.1 ± 1.92 | 2.91 ± 2.95 | 2.16 ± 2.2 | 0.08 |
| eGFR at study onset (ml/min) | 52.1 ± 29.7 | 52 ± 31.2 | 52.1 ± 29.3 | 0.78 |
| eGFR at last FU (ml/min) | 54 ± 33.7 | 51.3 ± 37.3 | 56 ± 32.6 | 0.11 |
| Follow-up time (y) | 2.5 (0.8–2.6) | 2.5 (0.9–2.6) | 2.5 (0.8–2.5) | 0.44 |
| **Comorbidities (n,%)** | | | | |
| • HBV | 37 (8.2%) | 9 (8.7%) | 28 (8.1%) | 0.85 |
| • HCV | 22 (4.9%) | 8 (7.7%) | 14 (4%) | 0.13 |
| • HIV | 2 (0.4%) | 1 (1%) | 1 (0.3%) | 0.4 |
| • Vascular disease | | | | |
| • Coronary artery disease | 35 (7.8%) | 10 (9.6%) | 25 (7.2%) | 0.46 |
| • Cerebrovascular disease | 17 (3.8%) | 2 (1.9%) | 15 (4.3%) | |
| • Peripheric vascular disease | 9 (2%) | 1 (1%) | 8 (2.3%) | |
| • Heart failure | 72 (16%) | 16 (15.4%) | 56 (16.1%) | 0.85 |
| • Diabetes mellitus | 72 (16%) | 17 (16.3%) | 55 (15.9%) | 0.9 |
| • Diabetic retinopathy | 35 (7.8%) | 5 (4.8%) | 30 (8.6%) | 0.3 |
| • Diabetic neuropathy | 5 (1.1%) | 2 (1.9%) | 3 (0.9%) | 0.3 |
| • HTA | 339 (75.2%) | 76 (73.1%) | 263 (75.8%) | 0.57 |
| • Obesity | 84 (18.6%) | 19 (18.3%) | 65 (18.7%) | 0.9 |
| • Solid malignancy | 20 (4.4%) | 5 (4.8%) | 15 (4.3%) | 0.79 |
| • Hematologic malignancy | 23 (5.1%) | 6 (5.8%) | 17 (4.9%) | 0.72 |
| **Treatment** | | | | |
| ACEI/ARBs | 346 (76.7%) | 80 (76.9%) | 266 (76.7%) | 0.95 |
| Immunosuppressive agents | 324 (71.8%) | 86 (82.7%) | 238 (68.6%) | 0.005 |
| • Steroids | 270 (59.9%) | 78 (75%) | 192 (55.3%) | <0.001 |
| • Cyclophosphamide | 138 (30.6%) | 44 (42.3%) | 94 (27.1%) | 0.003 |
| • MMF | 92 (20.4%) | 32 (30.8%) | 60 (17.3%) | 0.003 |
| • CNIs | 49 (10.9%) | 13 (12.5%) | 36 (10.4%) | 0.54 |
| • Rituximab | 81 (18%) | 27 (26%) | 54 (15.6%) | 0.01 |
| • Azathioprine | 72 (16%) | 16 (15.4%) | 56 (16.1%) | 0.85 |

**Abbreviations**: y, years; SCr, serum creatinine; M, male; F, female; eGFR, estimated glomerular filtration rate; FU, follow-up; HBV, hepatitis B virus, HCV, hepatitis C virus; HTA, arterial hypertension; ACEI, angiotensin-converting enzyme inhibitor; ARBs, angiotensin receptor blockers; MMF, mycophenolate mofetil; CNIs, calcineurin inhibitors.

infection follow-up was 2.5 y (IQR: 1.8–2.5). Of these, 70 patients had a mild-moderate form of COVID-19, 31 patients a severe form and 3 patients a COVID-19 related-death. Nine patients were identified to have two SARS-CoV-2 infections during the study follow-up, with all the subsequent infections being mild in severity. Another 16 patients died during the study follow-up due to reasons unrelated to SARS-CoV-2 infection (12 patients due to cardiovascular causes, 3 patients due to underlying malignancy and 1 patient due to a non-SARS-CoV-2

infection). All patients that died (n = 19) during the study follow-up were excluded from the analysis of renal function evolution.

Patients with SARS-CoV-2 infection were younger and received more frequently IS therapy [steroids, cyclophosphamide, mycophenolate mofetil (MMF) and rituximab] compared to those without infection, but the renal function at baseline was similar between the study groups.

## Renal outcome in relation to SARS-CoV-2 infection

Overall, 17.6% of patients reached the primary study endpoint, of whom 12.3% progressed to ESRD. Those with SARS-CoV-2 infection were more likely to reach the composite endpoint compared to those without infection [prevalence of composite endpoint, 26.7% vs. 14.8%; odds ratio (OR), 2.1 (95%CI, 1.23–3.58), p = 0.006]. Similarly, there was a significant decline of eGFR in the first year of follow-up between the two study groups [-2.24 (95%CI, -4.86 to 0.37) vs. 2.31 (95%CI, 0.78 to 3.85) ml/min, respectively, p = 0.004] (Figs 1 and 2A). The adjusted mean difference of eGFR change in the first year was -4.68 ml/min (95%CI, -7.76 to -1.59, p = 0.03). The results remained consistent after excluding patients with early progression to ESRD (within 12 months from baseline, n = 22), with the most accentuated decline of eGFR being noticed at 12 months from study onset (Figs 1A and 2B).

When restricting the analysis to patients with a kidney biopsy at least 12 months prior to COVID-19 pandemic, there was a significant difference in eGFR change between those with SARS-CoV-2 infection and those without infection [eGFR change/y pre-COVID-19: 1.29 (95%CI, -3.09 to 5.67) vs. 0.05 (95%CI, -1.8 to 1.9) ml/min/y, p = 0.92; eGFR change/y post-COVID-19: -1.95 (95%CI, -4.24 to 0.33) vs. -1.63 (95%CI, -4.07 to 0.81) ml/min/y, p = 0.004] (Fig 1B, Table 2). Similar to the analysis of the entire cohort, the most important difference was in the eGFR change in the first year post-COVID-19 [-6.1 (95%CI, -9.61 to -2.71) ml/min vs. +0.07 (95%CI, -1.67 to 1.8) ml/min, p<0.001], with an adjusted mean difference of -6.24 ml/min (95%CI, -9.87 to -2.6) (Table 2).

Subgroup analyses according to IS therapy and severity of SARS-CoV-2 infection are presented in Table 2 and S2 Table. SARS-CoV-2 infection was associated with a greater eGFR decline and a worse renal survival irrespective of IS use, but patients without IS showed a greater total eGFR decline after the SARS-CoV-2 infection compared to those with IS or without infection (Table 2, Figs 2C, 2D and 3A–3C). Regarding the severity of infection, patients with severe forms of SARS-CoV-2 infection had the greatest eGFR change in the first year

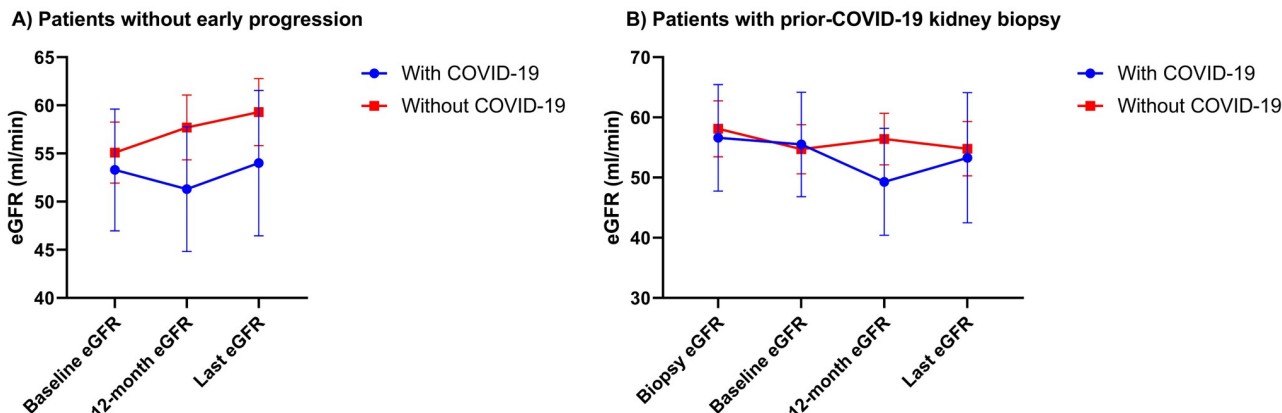

**Fig 1. eGFR evolution.** A) Patients without early progression to ESRD (<12 months). B) Patients with prior COVID-19 pandemic kidney biopsy.

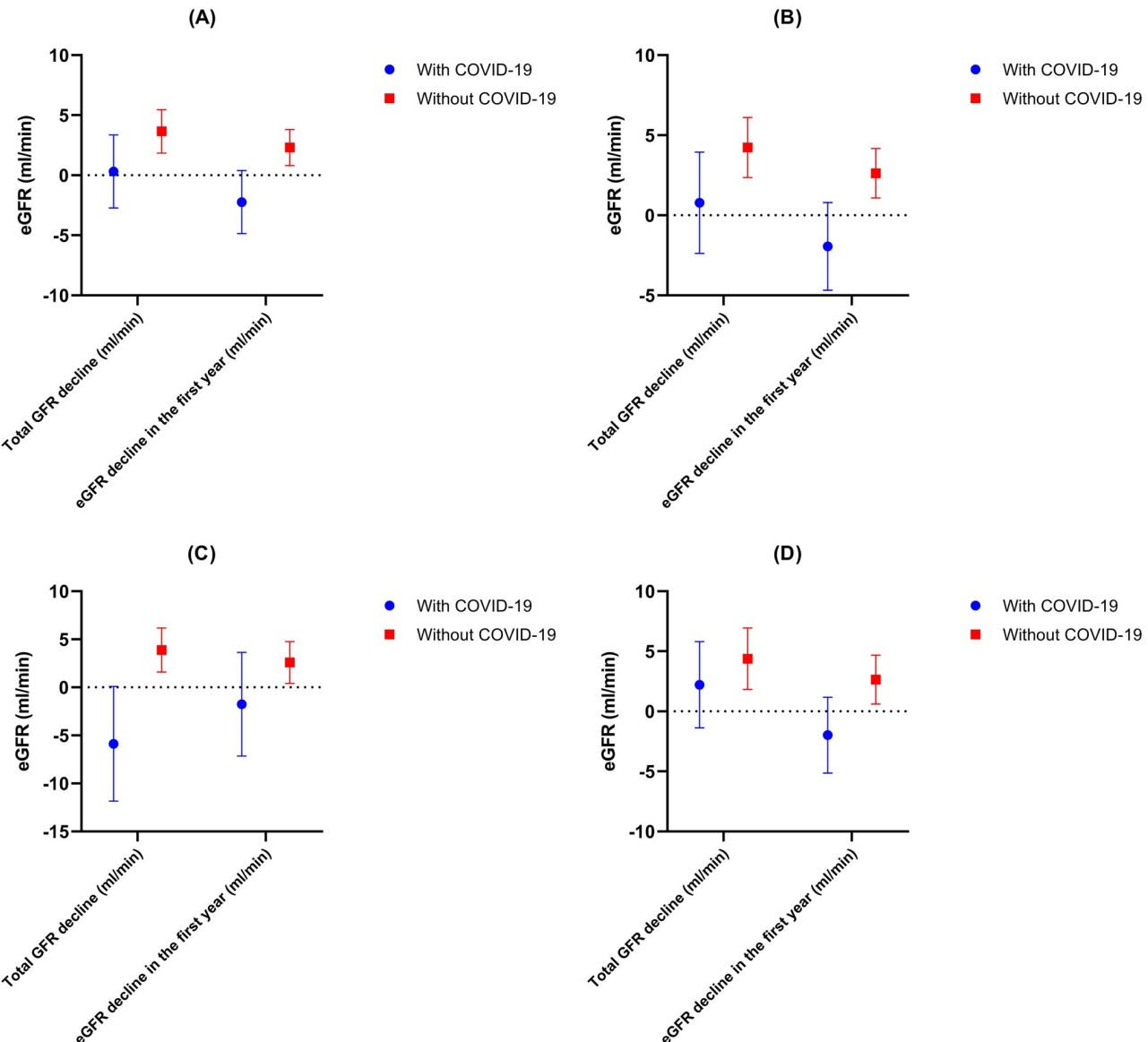

**(A)**

**(B)**

**(C)**

**(D)**

**Fig 2. eGFR decline.** A) Entire cohort. B) Patients without early progression to ESRD (<12 months). C) Patients without immunosuppression. D) Patients with immunosuppression.

[-5.12 (95%CI,-9.87 to -0.38) ml/min compared to -0.97 (95%CI, -4.14 to 2.2) ml/min in mild-moderate forms and to +2.31 (95%CI, 0.78 to 3.85) ml/min in those without infection, p = 0.005], the greatest eGFR change/y [-3.66 (95%CI, -7.04 to -0.28) ml/min/y compared to +0.93 (95%CI, -0.83 to 2.7) ml/min/y in mild-moderate forms and to -0.69 (95%CI, -2.81 to 3.38) in those without infection, respectively; p = 0.017] and the worst renal survival (proportion of patients with combined endpoint: 32.3% vs. 24.3% vs. 14.8%; p = 0.01)(Fig 3D). The mean adjusted difference of eGFR change in the first year was -7.81 ml/min [(95%CI, -14.04 to -1.59), p = 0.08; in severe forms versus no infection), -4.52 ml/min [(95%CI, -11.65 to 2.61), p = 0.38; in severe versus mild-moderate forms) and -3.29 ml/min [(95%CI, -7.65 to 1.05), p = 0.2; in mild-moderate forms versus no infection).

**Table 2. Renal outcomes in relation to COVID-19 infection.** (analysis after excluding patients that died during the study observation period, n = 19).

| Variable | Entire cohort | With COVID-19 | Without COVID-19 | p-value | Adjusted mean difference (95% CI) | p-value |
|---|---|---|---|---|---|---|
| Number of patients | 432 | 101 | 331 | - | - | - |
| eGFR at study onset (ml/min) | 52.7 ± 29.7 | 51.6 ± 31.3 | 53.1 ± 29.2 | 0.48 | - | - |
| eGFR at 12 months (ml/min) | 54.7 ± 31.4 | 49.3 ± 32.4 | 56.4 ± 31 | 0.03 | - | - |
| eGFR at last FU (ml/min) | 55.6 ± 33.9 | 51.9 ± 37.6 | 56.7 ± 32.7 | 0.12 | - | - |
| Total eGFR change (ml/min) | 2.87 (1.31 to 4.43) | 0.31 (-2.72 to 3.35) | 3.65 (2.01 to 5.73) | 0.07 | -3.34 (-7.03 to 0.34) | 0.07 |
| eGFR change in the first year (ml/min) | 1.22 (-0.1 to 2.55) | -2.24 (-4.86 to 0.37) | 2.31 (0.78 to 3.85) | 0.004 | -4.68 (-7.7 to -1.59) | 0.03 |
| eGFR change /y (ml/min/y) | -0.64 (-3.19 to 1.9) | -0.47 (-2.1 to 1.15) | -0.69 (-2.81 to 3.38) | 0.14 | 0.36 (-5.64 to 6.36) | 0.9 |
| eGFR decline >30% (%) | 14.6% | 23.8% | 11.8% | 0.003 | - | - |
| ESRD (%) | 12.3% | 21.8% | 9.4% | 0.001 | - | - |
| Combined endpoint (%) | 17.6% | 26.7% | 14.8% | 0.006 | - | - |
| **Patients without early progression to ESRD (< 12 months)** | | | | | | |
| Number of patients | 410 | 96 | 314 | - | - | - |
| eGFR at study onset (ml/min) | 54.6 ± 29.1 | 53.3 ± 31.2 | 55.1 ± 28.5 | 0.42 | - | - |
| eGFR at 12 months (ml/min) | 56.2 ± 30.8 | 51.3 ± 32 | 57.7 ± 30.3 | 0.06 | - | - |
| eGFR at last FU (ml/min) | 58.1 ± 32.9 | 54 ± 37.3 | 59.3 ± 31.4 | 0.09 | - | - |
| Total eGFR change (ml/min) | 3.42 (1.8 to 5.04) | 0.78 (-2.38 to 3.94) | 4.23 (2.34 to 6.11) | 0.09 | -3.5 (-7.3 to 0.32) | 0.07 |
| eGFR change in the first year (ml/min) | 1.55 (0.2 to 2.91) | -1.94 (-4.68 to 0.79) | 2.62 (1.08 to 4.17) | 0.008 | -4.68 (-7.8 to -1.52) | 0.004 |
| eGFR change /y (ml/min/y) | 1.7 (0.91 to 2.5) | 0.17 (-1.39 to 1.74) | 2.17 (1.25 to 3.09) | 0.1 | -2.03 (-3.9 to -0.16) | 0.03 |
| eGFR decline >30% (%) | 11% | 20.8% | 8% | <0.001 | - | - |
| ESRD (%) | 7.8% | 17.7% | 4.8% | <0.001 | - | - |
| Combined endpoint (%) | 13.4% | 22.9% | 10.5% | 0.002 | - | - |
| **Patients without immunosuppression** | | | | | | |
| Number of patients | 126 | 18 | 108 | - | - | - |
| eGFR at study onset (ml/min) | 54.7 ± 27.8 | 42.4 ± 24.5 | 56.8 ± 27.9 | 0.04 | - | - |
| eGFR at 12 months (ml/min) | 56.6 ± 30.3 | 40.6 ± 26.2 | 59.4 ± 30.2 | 0.01 | - | - |
| eGFR at last FU (ml/min) | 57.1 ± 31 | 36.5 ± 25.3 | 60.7 ± 30.7 | 0.003 | - | - |
| Total eGFR change (ml/min) | 2.46 (0.25 to 4.67) | -5.88 (-11.8 to 0.11) | 3.89 (1.6 to 6.2) | 0.001 | -9.66 (-15.7 to -3.53) | 0.002 |
| eGFR change in the first year (ml/min) | 1.95 (-0.06 to 3.97) | -1.76 (-7.19 to 3.66) | 2.59 (0.4 to 4.78) | 0.26 | -4.25 (-10.04 to 1.54) | 0.15 |
| eGFR change /y (ml/min/y) | 1.22 (0.13 to 2.3) | -2.95 (-6.01 to 0.1) | 1.94 (0.81 to 3.06) | 0.002 | -4.79 (-7.8 to -1.78) | 0.002 |
| eGFR decline >30% (%) | 12.7% | 16.7% | 12% | 0.7 | - | - |
| ESRD (%) | 11.1% | 11.1% | 11.1% | 0.99 | - | - |
| Combined endpoint (%) | 15.1% | 22.2% | 13.9% | 0.47 | - | - |
| **Patients with immunosuppression** | | | | | | |
| Number of patients | 306 | 83 | 223 | - | - | - |
| eGFR at study onset (ml/min) | 54.6 ± 29.7 | 55.6 ± 32.1 | 54.3 ± 28.8 | 0.92 | - | - |
| eGFR at 12 months (ml/min) | 56.1 ± 31 | 53.6 ± 32.8 | 56.9 ± 30.4 | 0.37 | - | - |
| eGFR at last FU (ml/min) | 58.4 ± 33.7 | 57.8 ± 38.5 | 58.6 ± 31.8 | 0.66 | - | - |
| Total eGFR change (ml/min) | 3.8 (1.7 to 5.89) | 2.21 (-1.37 to 5.8) | 4.38 (1.83 to 6.93) | 0.64 | -2.1 (-6.83 to 2.61) | 0.38 |
| eGFR change in the first year (ml/min) | 1.4 (-0.31 to 3.12) | -1.98 (-5.15 to 1.17) | 2.64 (0.61 to 4.67) | 0.02 | -4.52 (-8.33 to -0.71) | 0.02 |

(*Continued*)

**Table 2.** (Continued)

| Variable | Entire cohort | With COVID-19 | Without COVID-19 | p-value | Adjusted mean difference (95% CI) | p-value |
|---|---|---|---|---|---|---|
| eGFR change /y (ml/min/y) | 1.89 (0.87 to 2.92) | 0.84 (-0.93 to 2.62) | 2.28 (1.03 to 3.53) | 0.65 | -1.4 (-3.71 to 0.9) | 0.23 |
| eGFR decline >30% (%) | 15.4% | 25.3% | 11.7% | 0.003 | - | - |
| ESRD (%) | 12.7% | 24.1% | 8.5% | <0.001 | - | - |
| Combined endpoint (%) | 18.6% | 27.7% | 15.2% | 0.01 | - | - |
| Patients with kidney-biopsy prior to COVID-19 pandemic* | | | | | | |
| Number of patients | 248 | 57 | 191 | - | - | - |
| eGFR at biopsy (ml/min) | 57.8 ± 32.7 | 56.6 ± 33.3 | 58.1 ± 32.5 | 0.66 | - | - |
| eGFR at study onset (ml/min) | 54.9 ± 29.4 | 55.5 ± 32.7 | 54.7 ± 28.5 | 0.97 | - | - |
| eGFR at 12 months (ml/min) | 54.7 ± 30.9 | 49.3 ± 33.5 | 56.4 ± 30 | 0.11 | - | - |
| eGFR at last FU (ml/min) | 54.5 ± 33.9 | 53.3 ± 40.8 | 54.8 ± 31.7 | 0.49 | - | - |
| eGFR change /y pre-COVID-19 (ml/min) | 0.33 (-1.4 to 2.08) | 1.29 (-3.09 to 5.67) | 0.05 (-1.8 to 1.9) | 0.91 | 1.18 (-2.93 to 5.29) | 0.57 |
| eGFR change /y post-COVID-19 (ml/min) | -1.7 (-3.65 to 0.23) | -1.95 (-4.24 to 0.33) | -1.63 (-4.07 to 0.81) | 0.04 | -0.39 (-4.97 to 4.17) | 0.86 |
| Total eGFR change pre-COVID-19 (ml/min) | -2.23 (-5.11 to 0.64) | 0.84 (-6.35 to 8.03) | -3.1 (-6.24 to -0.06) | 0.88 | 3.86 (-2.82 to 10.5) | 0.25 |
| Total eGFR change post-COVID-19 (ml/min) | -0.37 (-2.16 to 1.42) | -2.16 (-6.54 to 2.23) | 0.16 (-1.7 to 2.1) | 0.03 | -2.35 (-6.59 to 1.87) | 0.27 |
| eGFR change in the first year post-COVID-19 (ml/min) | -1.41 (-2.99 to 0.17) | -6.1 (-9.61 to -2.71) | 0.07 (-1.67 to 1.8) | <0.001 | -6.24 (-9.87 to -2.6) | 0.001 |
| eGFR decline >30% (%) | 19% | 33.3% | 14.7% | 0.02 | - | - |
| ESRD (%) | 13.3% | 28.1% | 8.9% | <0.001 | - | - |
| Combined endpoint (%) | 20.6% | 33.3% | 16.8% | 0.007 | - | - |

*Patients with at least 12 months of follow-up prior to COVID-19 pandemic

**Abbreviations**: y, years; eGFR, estimated glomerular filtration rate; FU, follow-up; ESRD, end-stage renal disease.

## Renal outcome after SARS-CoV-2 infection according to underlying etiology

A subgroup analysis according to the underlying etiology was conducted in most prevalent histological patterns of glomerular injury (S3 Table). The trend for a worse renal outcome following a SARS-CoV-2 infection remained consistent in patients with IgAN, MN and focal and segmental glomerulosclerosis (FSGS), with the exception of those with LN. The prevalence of the primary composite endpoint was higher after SARS-CoV-2 infection in patients with IgAN (34.4% vs. 15.9%, p = 0.03) and FSGS (66.7% vs. 26.5%, p = 0.07). In addition, the adjusted mean difference of total eGFR change following COVID-19 was -1.72 ml/min (95% CI, -5.66 to 2.22; p = 0.38) in IgAN, -8.3 ml/min (95%CI, -19.6 to 2.91; p = 0.14) in MN and -16.3 (95%CI, -38.9 to 6.2; p = 0.15) in FSGS (S3 Table).

## Risk factors for CKD progression

Table 3 details the models of Cox proportional hazards regression analysis regarding predictive factors for renal events. After multivariate adjustment, eGFR at baseline (HR, 0.94 for each 1 ml/min/1.73m$^2$; 95%CI, 0.92–0.95), occurrence of SARS-CoV-2 infection (HR, 1.91; 95% CI,1.16–3.12) and male gender (HR, 1.64; 95%CI, 1.01–2.66) were identified as independent predictors of the primary composite endpoint (Fig 4, **model A1**). In model A2, a graded

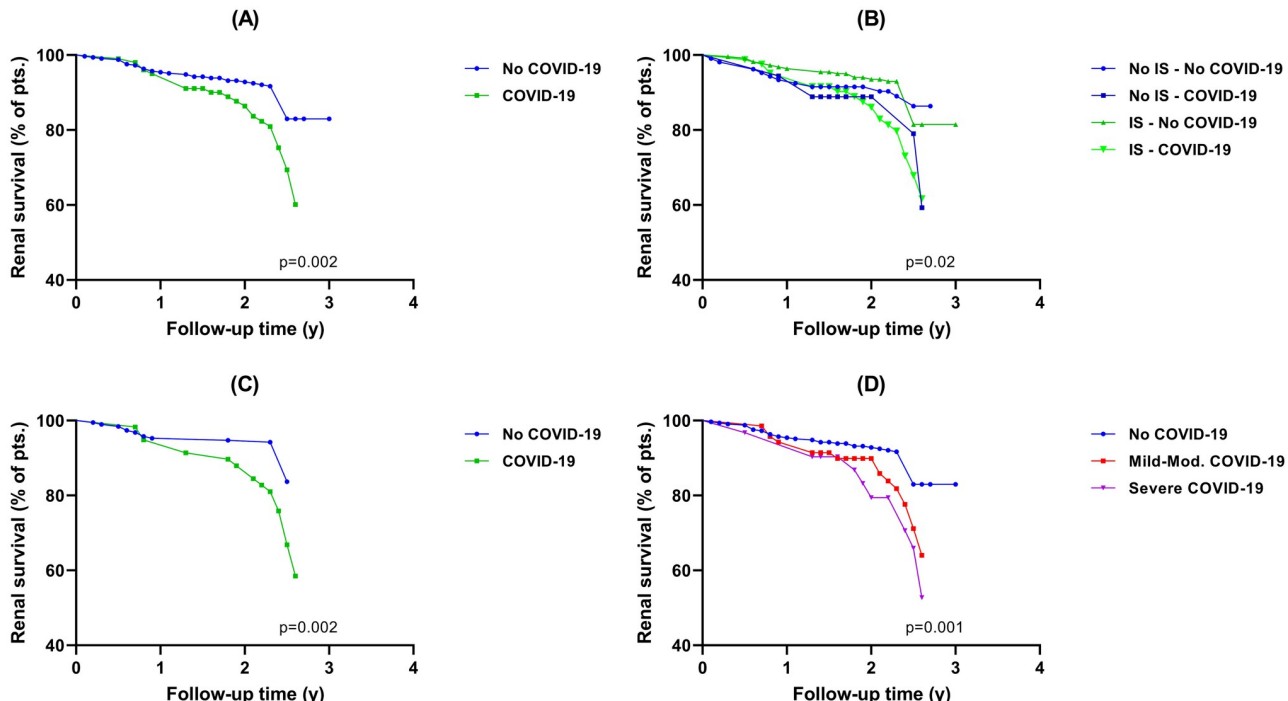

**Fig 3. Renal survival.** A) Entire cohort. B) Entire cohort, in relation to immunosuppression. C) Patients with pre-COVID-19 pandemic kidney biopsy. D) Entire cohort, in relation to the severity of COVID-19.

impact of SARS-CoV-2 infection on renal outcome was identified, with a mild-moderate and a severe infection increasing the risk for the composite endpoint by 1.8-fold and by 2.1-fold, respectively (Table 3, Fig 4–**model A2**). The results remained consistent for models with only ESRD as the outcome variable (Table 3, Fig 4–**model B1 and B2**).

In linear regression analysis with eGFR slope at 12 months as the outcome variable, the occurrence of SARS-CoV-2 infection independently determined a reduction of eGFR by 4.62 ml/min/1.73m$^2$ (β coefficient, -4.62; 95CI, -7.74 to -1.5; p = 0.004)(Table 4).

## Discussion

This study, the first to our knowledge to address the long-term impact of SARS-CoV-2 infection on renal function in patients with pre-existing biopsy-proven kidney diseases, confirms that COVID-19 is associated with accelerated eGFR decline and a worse renal survival, with a graded impact correlating with infection severity, those with severe forms of COVID-19 having the worst renal outcome.

Following the decline of COVID-19 pneumonia-related mortality, the focus regarding the infection with SARS-CoV-2 has now shifted to the long-term multi-organ functional impairment (e.g., pulmonary, cardiac and renal function, coagulopathy) [14, 15]. Accumulating data supports an increased risk for renal events following SARS-CoV-2 infection, including AKI and future renal function decline, new-onset or relapsing glomerular disorders [5, 7, 8, 16–19]. Nonetheless, previous studies are characterized by a significant heterogeneity in terms of the study population included, while reports on patients with pre-existing CKD are scarce [7]. Accordingly, we have selected a cohort of patients from a renal biopsy registry with pre-existing CKD and adequate follow-up time to assess the impact of SARS-CoV-2 infection on

**Table 3. Cox proportional regression analysis regarding predictive factor of renal events.**

| Variable | Univariate analysis | | Multivariate analysis (Model 1) | | Multivariate analysis (Model 2) | |
|---|---|---|---|---|---|---|
| | Hazard ratio (95%CI) | p value | Hazard ratio (95%CI) | p value | Hazard ratio (95%CI) | p value |
| **Model A (combined endpoint)** | | | | | | |
| eGFR at baseline (for 1 ml/min/1.73m$^2$) | 0.94 (0.92–0.95) | <0.001 | 0.94 (0.92–0.95) | <0.001 | 0.94 (0.92–0.95) | <0.001 |
| COVID-19 (yes vs. no) | 1.93 (1.2–3.12) | 0.006 | 1.91 (1.16–3.12) | 0.01 | - | - |
| COVID-19 (vs. no COVID-19) | - | - | - | - | - | - |
| • Mild-moderate COVID-19 | 1.76 (1.008–3.068) | 0.04 | - | - | 1.8 (1–3.25) | 0.05 |
| • Severe COVID-19 | 2.34 (1.18–4.66) | 0.01 | - | - | 2.1 (1.02–4.3) | 0.04 |
| Immunosuppression (yes vs. no) | 1.15 (0.68–1.94) | 0.58 | 1.29 (0.74–2.25) | 0.54 | 1.29 (0.74–2.26) | 0.36 |
| Gender (males vs. female) | 1.42 (0.89–2.25) | 0.13 | 1.64 (1.01–2.66) | 0.04 | 1.68 (1.01–2.8) | 0.04 |
| Charlson Comorbidity Score | 1.25 (1.14–1.37) | <0.001 | 1.008 (0.89–1.13) | 0.88 | 1.005 (0.89–1.13) | 0.94 |
| Etiology (IgAN vs. other) | 1.23 (0.75–2.02) | 0.41 | 0.72 (0.4–1.31) | 0.29 | 0.73 (0.4–1.3) | 0.29 |
| **Model B (ESRD)** | | | | | | |
| eGFR at baseline (for 1 ml/min/1.73m$^2$) | 0.86 (0.83–0.89) | <0.001 | 0.85 (0.82–0.88) | 0.004 | 0.85 (0.82–0.88) | <0.001 |
| COVID-19 (yes vs. no) | 2.5 (1.45–4.33) | 0.001 | 2.32 (1.3–4.12) | 0.004 | - | - |
| COVID-19 (vs. no COVID-19) | - | - | - | - | - | - |
| • Mild-moderate COVID-19 | 2.3 (1.22–4.33) | 0.01 | - | - | 2.16 (1.05–4.43) | 0.03 |
| • Severe COVID-19 | 2.97 (1.36–6.47) | 0.006 | - | - | 2.56 (1.13–5.75) | 0.02 |
| Immunosuppression (yes vs. no) | 1.08 (0.58–1.99) | 0.79 | - | | - | |
| Gender (males vs. female) | 1.35 (0.78–2.35) | 0.27 | 1.89 (1.07–3.31) | 0.02 | 1.97 (1.06–3.64) | 0.03 |
| Charlson Comorbidity Score | 1.34 (1.2–1.49) | <0.001 | 1.13 (0.99–1.28) | 0.05 | 1.12 (0.98–1.28) | 0.08 |
| Etiology (IgAN vs. other) | 1.44 (0.81–2.56) | 0.21 | - | | - | |

**Abbreviations**: y, years; eGFR, estimated glomerular filtration rate; ESRD, end-stage renal disease; IgAN, IgA nephropathy; ACEI, angiotensin-converting enzyme inhibitor; ARB, angiotensin receptor blocker.

long-term renal function. Our study cohort differs from previous cohorts in that it consists mainly of glomerular disorders (84%), with a distribution of the underlying pathology that is consistent with that reported in the European population [20, 21]. By comparison, Schmidt-Lauber *et al* compared the kidney outcomes after mild-moderate COVID-19 in 443 patients with 1328 matched-controls from the general population without prior COVID-19 [10]. In this study, the mean eGFR was only slightly lower after a median 9 months following a non-severe COVID-19 and there were no signals suggesting a risk for progressive kidney dysfunction [10]. Nonetheless, the study cohort included only approximately 10% of patients with pre-existing CKD, among whom only several patients had an underlying etiological diagnosis [10]. Similarly, Bowe *et al* reported kidney outcomes following COVID-19 in a population-based study that included US veterans and identified an increased risk of AKI, eGFR decline and ESRD after a median follow-up period of approximately 6 months [7]. The mean eGFR in this study was over 70 ml/min/1.73m$^2$, while the proportion of patients with CKD is unknown [7] By comparison, our study included patients with a biopsy-proven etiological diagnosis at higher risk for CKD progression, with a significantly lower mean eGFR (52.7 ± 29.7 ml/min/1.73m$^2$) and a longer follow-up period (median 2.5 years). Thus, our study setting might be more accurate to evaluate whether COVID-19 is a risk factor for CKD progression.

We have identified a significant difference of eGFR change in the first year following a SARS-CoV-2 infection with a mean adjusted difference of -4.68 ml/min (95%CI, -7.7 to 1.59), which is higher than the -1.8 ml/min (95%CI, -3.2 to -0.5) reported in the study by Schmidt-Lauber et al, albeit after a lower follow-up period (median, 9 months) [10]. Nonetheless, that

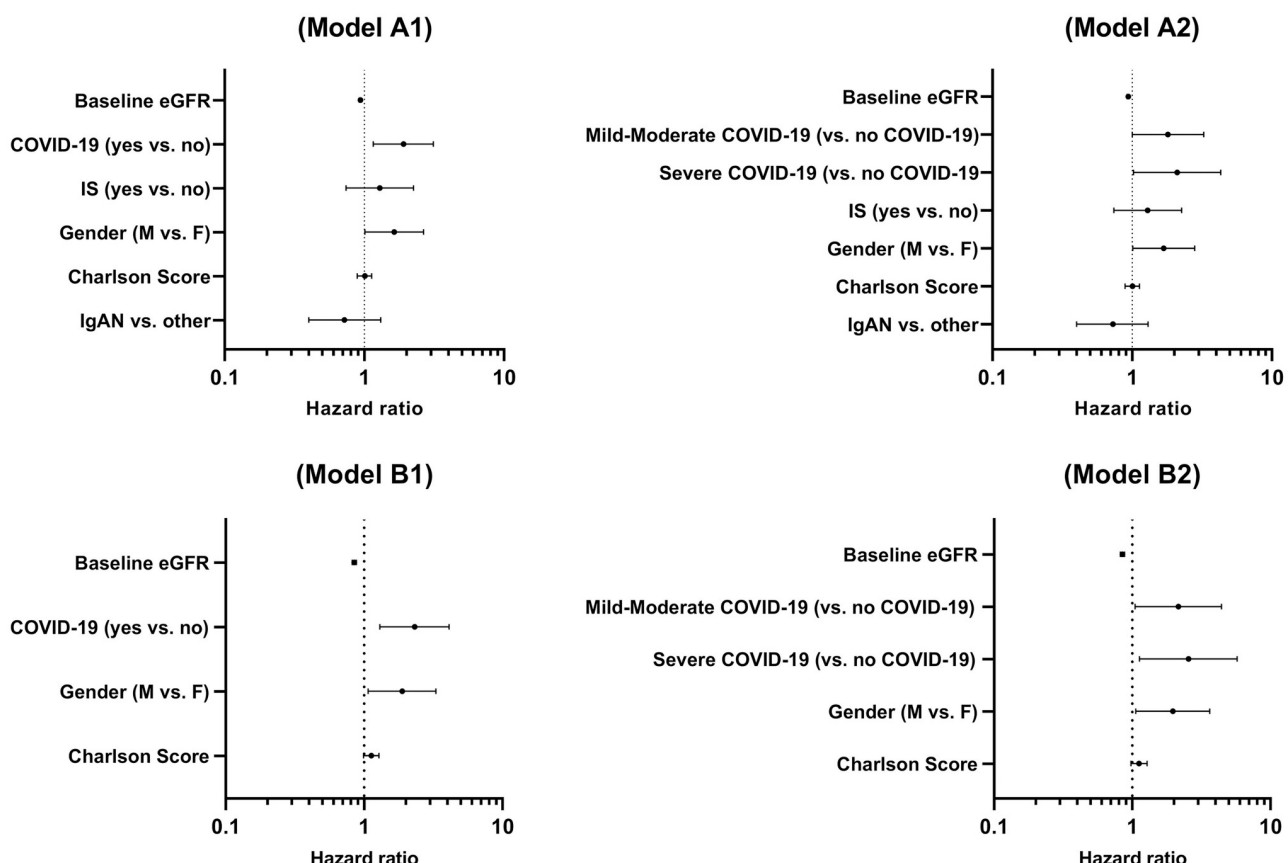

**Fig 4. Predictive factors of renal outcome.** A1, A2) Combined endpoint. B1, B2) ESRD.

study cohort included patients with fewer comorbidities and excluded patients with severe forms of COVID-19 [10]. Indeed, in our cohort those with a severe infection had the worst renal outcome (with an adjusted mean difference of eGFR change at 12 months of -7.8 ml/min compared to those without infection). However, despite not reaching a statistical significance,

**Table 4. Linear regression analysis regarding predictors of eGFR at 12 months after COVID-19 pandemic onset.**

| Variables | Unadjusted | | | Adjusted* | | |
|---|---|---|---|---|---|---|
| | β coefficient (95%CI) | Std. error | p-value | β coefficient (95%CI) | Std. error | p-value |
| eGFR at baseline (for 1 ml/min/1.73m²) | -0.05 (-0.1; -0.01) | 0.02 | 0.01 | -0.1 (-0.15; -0.05) | 0.02 | <0.001 |
| COVID-19 (yes vs. no) | -4.57 (-7.75; -1.4) | 1.61 | 0.005 | -4.62 (-7.74; -1.5) | 1.58 | 0.004 |
| Immunosuppression (yes vs. no) | -0.55 (-3.56; 2.4) | 1.53 | 0.71 | 0.28 (-2.6; 3.2) | 1.51 | 0.85 |
| ACEI/ARB (yes vs. no) | -1.86; (-5.1; 1.4) | 1.67 | 0.26 | - | - | - |
| Age at baseline (for 1 y) | -0.52 (-0.14; 0.04) | 0.04 | 0.26 | - | - | - |
| Gender (males vs. female) | -3.79 (-6.48; -1.1) | 1.36 | 0.006 | -3.53 (-6.2; -0.87) | 1.35 | 0.009 |
| Charlson Comorbidity Score | -0.62 (-1.29; 0.03) | 0.34 | 0.06 | -1.37 (-2.1; -0.6) | 0.37 | <0.001 |
| Etiology (IgAN vs. other) | -1.68 (-4.86; 1.49) | 1.61 | 0.29 | -2.78 (-6.1; 0.53) | 1.68 | 0.09 |

**Abbreviations**: y, years; eGFR, estimated glomerular filtration rate; IgAN, IgA nephropathy; ACEI, angiotensin-converting enzyme inhibitor; ARB, angiotensin receptor blocker.

those with mild-moderate forms had a greater eGFR change at 12 months compared to those without COVID-19 (adjusted mean difference of -3.29 ml/min versus -1.8 ml/min in the previous study) [10]. This further supports the fact that our cohort included patients with a higher risk of CKD progression that may be more prone to worse renal survival following COVID-19. In addition, these results must be interpreted in the context of the treatment background, with more than 70% of patients having received various IS regimens. In this instance, the relation between IS therapy and COVID-19 is bidirectional. While COVID-19 might lead to a reduction in the intensity of the IS regimens, the continuation of IS may protect against further reactivation of the underlying glomerular disease and may mitigate to systemic inflammatory response associated with COVID-19 [22, 23]. On the other hand, the expected amelioration, or even improvement, in renal function following IS therapy in glomerular disorders may be blunted by the occurrence of SARS-CoV-2 infection. This was evident in our cohort in patients with IgAN, MN or FSGS, in whom the SARS-CoV-2 infection triggered a more rapid progression of the underlying disease [24]. Nonetheless, in multivariate linear regression analysis, the SARS-CoV-2 infection independently determined a reduction of eGFR by 4.62 ml/min at 12 months, which is close to the value of 5 ml/min/y that is stated in the 2012 KDIGO guidelines for the management of CKD as a threshold to define a rapid progression [25].

The longer follow-up period of this study compared to previous reports further highlighted another important aspect. The SARS-CoV-2 impact on eGFR change was most evident after one year with the effect attenuating over the following period of observation. Similarly, a study that included 2,212 long-COVID patients referred to post-COVID recovery clinics showed that there was an estimated 2.96 ml/min/1.73m$^2$ decreased in eGFR within 1 year after COVID-19, with more than 40% of patients being at risk of CKD [26]. While the study was not designed to address this aspect, it appears that SARS-CoV-2 infection may have induced an injury resembling AKI. The relation between SARS-CoV-2 infection and AKI has been well established, with both virus-related and virus-unrelated effects (e.g., related to sepsis, nephrotoxic agents, cardiovascular instability, etc.) being proposed to account for the renal injury [11]. Although a definitive proof for the renal tropism of SARS-CoV-2 is lacking [11, 27], recent data shows that SARS-CoV-2 infected human-induced pluripotent stem-cell derived kidney organoids were characterized by activation of profibrotic signaling pathways [28], while other studies have suggested that the inflammatory burden translated into worse outcomes following COVID-19 [29]. This might further support the direct negative impact of COVID-19 on renal function. Although we have noticed an amelioration of eGFR decline after the first year, it is well known that episodes of AKI are a significant risk factor for both CKD progression (in those with pre-existing CKD) and future CKD development (in those with a previous normal renal function) [30]. However, our results should be regarded only as a hypothesis with respect to the relation between SARS-CoV-2 infection and AKI, as a definitive proof for this aspect is lacking due to the absence of AKI assessment episodes and the inability to evaluate for AKI biomarkers. A recent report from the COV-GN registry identified a prevalence of 16.9% of AKI among patients with glomerular disorders and SARS-CoV-2 infection (n = 59), while those with severe COVID-19 being more likely to develop AKI [31]. Nonetheless, our hypothesis is further supported in this study by the observation that SARS-CoV-2 infection is independently associated with an almost 2-fold higher risk for the primary composite endpoint (persistent decline in eGFR of more than 30% or ESRD) and a 2.3-fold higher risk for ESRD, while this effect may become even more pronounced with a prolongation of the follow-up period. Moreover, in patients with kidney biopsy prior to COVID-19 pandemic, we identified a significant difference of eGFR change/y post-COVID-19 [-1.95 ml/min/y (95%CI, -4.24 to 0.33)] compared to the pre-COVID-19 period [1.29 ml/min/y (95%CI, -3.09 to 5.67)]. A recent analysis from the

CureGN cohort, that included 2,055 patients with glomerular disorders (IgAN, FSGS, MN and minimal-change disease) showed that, despite not reaching statistical significance, the post-SARS-CoV-2 renal function decline was more accelerate compared to pre-pandemic period (-4.26 ml/min/1.73 m$^2$/y versus -1.4 ml/min/1.73 m$^2$/y) [32].

This study has several limitations that need to be acknowledged. First, this is a single-center, retrospective, observational study, and these findings need to be validated in other populations. In addition, the retrospective nature of the study might introduce potential confounding factors other than those adjusted for in multivariate analysis. Moreover, given the variability in IS agents used among glomerular disorders, we could not evaluate distinctly every IS regimen. However, this is the first study to stratify renal outcomes post-SARS-CoV-2 infection according to the use of IS agents and showed that COVID-19 occurrence influenced renal function decline irrespective of the use of IS therapy, albeit to a higher degree in those without IS therapy exposure. Second, we need to take into account that although the study cohort included mostly glomerular disorders, there might be significant differences in renal function decline according to the underlying etiology (e.g., ANCA-associated vasculitis versus minimal-change disease), as noted in the analysis limited to the most prevalent glomerular diseases. Thus, the heterogeneity of the underlying pathology might bring a difficulty in the evaluation of renal disease progression. However, as opposed to other population-based studies in which the CKD was defined solely on the eGFR and/or albuminuria assessment, our attempt to account for heterogeneity of the underlying disease and of the IS therapy provides a better characterization of the study population. Nonetheless, given the frequent use of IS therapy in this cohort (over 70% of patients), and the known impact of treatment on controlling glomerular disease activity, it is likely that the accelerated renal function decline may be related to COVID-19. Third, we could not account for the impact of proteinuria or urinalysis (hematuria, leukocyturia) on renal outcome due to the retrospective nature of the study and lack of assessment in all patients. However, this is the first cohort of patients with biopsy-proven pre-existing kidney diseases evaluated for the impact of SARS-CoV-2 infection on renal survival, with a significantly higher risk of renal function decline and a longer follow-up period compared to previous reports.

In conclusion, we have identified a significant impact of SARS-CoV-2 infection on long-term renal function in patients with biopsy-proven kidney diseases that leads to a greater decline of eGFR and a worse renal survival, those with severe forms of COVID-19 having the worst renal outcome.

## Supporting information

**S1 Fig. Study flow-chart.**
(TIF)

**S2 Fig. Etiology of renal disease.**
(TIF)

**S1 Table. Etiology of renal disease.**
(DOCX)

**S2 Table. Sub-analysis according to the severity of COVID-19.**
(DOCX)

**S3 Table. Renal function outcomes in relation to underlying etiology.**
(DOCX)

## Author Contributions

**Conceptualization:** Bogdan Obrişcă, Roxana Jurubiţă, Bogdan Sorohan, Camelia Achim, Andreea Andronesi, Georgia Micu, Gener Ismail.

**Data curation:** Bogdan Obrişcă, Valentin Mocanu, Alexandra Vornicu, Roxana Jurubiţă, Bogdan Sorohan, George Dimofte, Camelia Achim, Andreea Andronesi, Georgia Micu, Raluca Bobeică, Nicu Caceaune, Alexandru Procop, Vlad Herlea, Mihaela Gherghiceanu, Gener Ismail.

**Formal analysis:** Bogdan Obrişcă, Bogdan Sorohan, Andreea Andronesi, Gener Ismail.

**Investigation:** Bogdan Obrişcă, Valentin Mocanu, Alexandra Vornicu, Roxana Jurubiţă, Bogdan Sorohan, George Dimofte, Camelia Achim, Andreea Andronesi, Georgia Micu, Raluca Bobeică, Nicu Caceaune, Alexandru Procop, Vlad Herlea, Mihaela Gherghiceanu, Gener Ismail.

**Methodology:** Bogdan Obrişcă, Bogdan Sorohan, Camelia Achim, Andreea Andronesi, Georgia Micu, Gener Ismail.

**Supervision:** Gener Ismail.

**Writing – original draft:** Bogdan Obrişcă, Gener Ismail.

**Writing – review & editing:** Bogdan Obrişcă, Valentin Mocanu, Alexandra Vornicu, Roxana Jurubiţă, Bogdan Sorohan, George Dimofte, Camelia Achim, Andreea Andronesi, Georgia Micu, Raluca Bobeică, Nicu Caceaune, Alexandru Procop, Vlad Herlea, Mihaela Gherghiceanu, Gener Ismail.

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
