## [Decision Letter · Decision Letter 0]

2 Oct 2023

PONE-D-23-25206The impact of SARS-CoV-2 infection on renal function in patients with biopsy-proven kidney diseasesPLOS ONE

Dear Dr. Obrisca,

Thank you for submitting your manuscript to PLOS ONE. After careful consideration, we feel that it has merit but does not fully meet PLOS ONE’s publication criteria as it currently stands. Therefore, we invite you to submit a revised version of the manuscript that addresses the points raised during the review process.

**The manuscript addresses a topic of potential interest; however, there are significant drawbacks in the study that need to be addressed to draw robust conclusions. To highlight some of these limitations, i) acknowledgement that the retrospective nature of the study introduces several confounding factors and raises questions about whether renal progression may be attributed to underlying renal diseases *per se*; ii) lack of indications on the proportion of patients in the positive group who discontinued therapy, and variations in immunosuppressive protocols could impact renal progression; iii) no information is provided regarding the systematic COVID-19 nucleic acid screening during follow-up time in the negative group; iv) absence of statistical evaluation of differences in 24-hour urinary protein between negative and positive patients during the follow-up time; iv) the number of endpoint events in COX regression should be at least 10 times greater than the number of independent variables, while here ESRD only occurred more than 30 times, so the independent variables of COX regression should be limited only to 3 to 4 terms.**

We look forward to receiving your revised manuscript.

Kind regards,

Giuseppe Remuzzi

Academic Editor

PLOS ONE

3. We notice that your supplementary figures are uploaded with the file type 'Figure'. Please amend the file type to 'Supporting Information'. Please ensure that each Supporting Information file has a legend listed in the manuscript after the references list.

4. Please include captions for your Supporting Information files (Supplemental Tables) at the end of your manuscript, and update any in-text citations to match accordingly. Please see our Supporting Information guidelines for more information: http://journals.plos.org/plosone/s/supporting-information.

Reviewers' comments:

Reviewer's Responses to Questions

**Comments to the Author**

1. Is the manuscript technically sound, and do the data support the conclusions?

Reviewer #1: Partly

Reviewer #2: Partly

2. Has the statistical analysis been performed appropriately and rigorously? 

Reviewer #1: Yes

Reviewer #2: Yes

3. Have the authors made all data underlying the findings in their manuscript fully available?

Reviewer #1: Yes

Reviewer #2: Yes

4. Is the manuscript presented in an intelligible fashion and written in standard English?

Reviewer #1: Yes

Reviewer #2: Yes

5. Review Comments to the Author

Reviewer #1: The idea of the paper is interesting however I have some comments.

1. The study is retroepective which made many confounding factors to involve.The renal progression may be caused by underlying renal diseases per se .The different IS protocols may have impact on the renal progression,also any acute illness or other medications can have impact.How can the author explain this .

2.It appears that SARS-CoV-2 infection may have induced a form of AKI and even many sentences that specify the effect of SARS-COV-2 and kidney that implies to this study should have more proof and state in the manuscript.

Reviewer #2: 1、Was COVID-19 nucleic acid screening regularly performed during follow-up time (For example, every 2 weeks or at least 1 month) in the negative group. And how to determine whether the patients in the negative group were never infected with COVID-19 during the follow-up time?

2. Please describe the number and cause of death of patients in the two groups respectively.

3. Please describe the proportion of patients in the positive group who discontinued immunosuppressive therapy.

4. Please describe the number of COVID-19 infections for each patient in the positive group.

5. Were there significant differences in 24-hour urinary protein between the two groups during the follow-up time?

6. The number of endpoint events in COX regression should be at least 10 times greater than the number of independent variables. In this paper, ESRD only occurred more than 30 times, so the independent variables of COX regression should be limited only to 3 to 4 terms.

6. PLOS authors have the option to publish the peer review history of their article (what does this mean?). If published, this will include your full peer review and any attached files.

Reviewer #1: No

Reviewer #2: No

---

## [Author Response · Author response to Decision Letter 0]

13 Oct 2023

Dear Plos One Editorial team, 

On behalf of the co-authors, I want to thank you for the opportunity of incorporating editorial and reviewer comments made in relation to our manuscript entitled “The impact of SARS-CoV-2 infection on renal function in patients with biopsy-proven kidney diseases”. We hope to have addressed all the comments and suggestions and believe that it has made our report clearer and more meaningful for publication. 

Sincerely, 

Bogdan Obrisca

Editor

The manuscript addresses a topic of potential interest; however, there are significant drawbacks in the study that need to be addressed to draw robust conclusions. To highlight some of these limitations, i) acknowledgement that the retrospective nature of the study introduces several confounding factors and raises questions about whether renal progression may be attributed to underlying renal diseases per se; ii) lack of indications on the proportion of patients in the positive group who discontinued therapy, and variations in immunosuppressive protocols could impact renal progression; iii) no information is provided regarding the systematic COVID-19 nucleic acid screening during follow-up time in the negative group; iv) absence of statistical evaluation of differences in 24-hour urinary protein between negative and positive patients during the follow-up time; iv) the number of endpoint events in COX regression should be at least 10 times greater than the number of independent variables, while here ESRD only occurred more than 30 times, so the independent variables of COX regression should be limited only to 3 to 4 terms.

Response: We appreciate the suggestions made by the editor and reviewers in relation to our manuscript. We hope that based on these suggestions, our revised manuscript is clearer and our results more meaningful. In addition, during the review process of this manuscript, two other studies regarding the relation of glomerular disorders with SARS-CoV-2 have been published and the reference list has been updated accordingly (Wang C et al, Association of COVID-19 Versus COVID-19 Vaccination With Kidney Function and Disease Activity in Primary Glomerular Disease: A Report of the Cure Glomerulonephropathy Study, Am J Kid Dis; Gauckler P et al, COVID-19 outcomes in patients with a history of immune mediated glomerular diseases, Frontiers in Immunology). However, as the CureGN cohort only includes patients with IgAN/MCD/FSGS/MN and the second study included only 59 patients with glomerular disorders and SARS-CoV-2 infection, we appreciate that our study included a cohort with a variety of patients with glomerular disorders that is representative of a real-world renal biopsy registry from a tertiary center and with a longer post-infection follow-up compared to previous studies (median 2.5 years compared to 0.8 years in the CureGN cohort). 

Nonetheless, our study has several limitations as pointed out by the reviewers that need to be acknowledged. Accordingly, our study limitation section has been revised appropriately. 

i) acknowledgement that the retrospective nature of the study introduces several confounding factors and raises questions about whether renal progression may be attributed to underlying renal diseases per se;

Response: We agree that the retrospective nature of the study might introduce potential confounding factors other than those adjusted for in the multivariate analysis, and we updated the limitations of the study section accordingly. However, in an attempt to overcome this aspect, we stratified the analysis according to immunosuppression use, type of underlying disorders (for the most frequent etiologies) and adjust in Cox regression for the confounding effect of IS and type of glomerular disorder on renal outcome. Moreover, given the widespread use of IS in this cohort (over 70%) and the known relation between treatment and glomerular disease activity, it is likely that the worsening of renal function may be related to COVID-19, irrespective of the underlying etiology. 

ii) lack of indications on the proportion of patients in the positive group who discontinued therapy, and variations in immunosuppressive protocols could impact renal progression.

Response: Our center’s protocol was not to modify the immunosuppressive regimens during the COVID-19 pandemic. However, during a SARS-CoV-2 infection a temporarily reduction in the doses of immunosuppressive regimens (for up to 10-14 days following the resolution of the infection) was undertaken. We have mentioned this aspect in the methods section. Regarding the second part of the question, we agree that there is a significant heterogeneity of immunosuppression regimens used among different glomerular disorders. However, it is difficult to adequately evaluate the independent effect of each IS regimen. Given this limitation (that is encountered in any study enrolling patients from a biopsy registry), we attempted to stratify the analysis according to the use of immunosuppression showing that COVID-19 occurrence influenced renal function decline irrespective of immunosuppression use, albeit to a higher degree in those without immunosuppression exposure. We have also highlighted this aspect in the discussion. 

iii) no information is provided regarding the systematic COVID-19 nucleic acid screening during follow-up time in the negative group.

Response: SARS-COV-2 infection screening was undertaken regularly in our center in this category of patients with a periodicity of 1-3 months. Our center’s protocol was to undertake a proactive screening for SARS-CoV-2 infection in this category of immunosuppressed patients via outpatient visits, hospital admissions, periodic telephone interviews and a thorough review of the electronic clinical health records during the study follow-up.

We have updated the methods section to describe this aspect. 

iv) absence of statistical evaluation of differences in 24-hour urinary protein between negative and positive patients during the follow-up time.

Response: Indeed, proteinuria is one of the most significant risk factors for CKD progression irrespective of the underlying etiology. However, because of the retrospective nature of this study and the initial mobility restriction in the context of the pandemic, we had many missing proteinuria assessments. We acknowledge that this is a limitation of the study and updated this section accordingly. In addition, in a recent CureGN analysis (published during the review of this paper), the impact of SARS-CoV-2 of future GFR loss remained meaningful even after adjustment for baseline proteinuria. So, with this drawback, we still believe that the impact of SARS-CoV-2 on renal outcome would have remained significant even after proteinuria inclusion in the model. Moreover, given the suggestion to limit the number of variables in the Cox model, we would still have a limited number of variables to potentially adjust for. In addition, the relevance of proteinuria on renal outcomes may differ significantly between different glomerular disorders (e.g.: IgAN vs. MCD vs. MN).

v) the number of endpoint events in COX regression should be at least 10 times greater than the number of independent variables, while here ESRD only occurred more than 30 times, so the independent variables of COX regression should be limited only to 3 to 4 terms.

Response: We appreciate this suggestion. In this cohort, we had 53 cases of ESRD and 76 cases of the composite endpoint (ESRD and more than 30% decrease in eGFR). Accordingly, in order to select the best prediction models, we modified our multivariate Cox regression models and chose a stepwise backward elimination method. According to this model, we were left with 6 variables in the analysis regarding the composite endpoint and 4 variables in the analysis regarding only ESRD. We have modified throughout the manuscript this new analysis and updated the methods section to highlight the method used for multivariate Cox regression.

Reviewer 1

The idea of the paper is interesting however I have some comments.

1. The study is retrospective which made many confounding factors to involve. The renal progression may be caused by underlying renal diseases per se. The different IS protocols may have impact on the renal progression, also any acute illness or other medications can have impact. How can the author explain this.

Response: We agree that the retrospective nature of the study might introduce potential confounding factors other than those adjusted for in the multivariate analysis, and we updated the limitations of the study section accordingly. We have tried to account for underlying disease in two ways. Firstly, by also analyzing the renal progression in the most frequent etiology categories. Second, we have adjusted in the multivariate analysis (Cox and linear regression) by the potential impact of the underlying etiology and use of immunosuppression. By this approach, we have identified that the impact of SARS-CoV-2 infection on renal function decline occurred independently of the underlying etiology and was most evident in patients with IgAN, FSGS and MN. In addition, we agree that given the variability of immunosuppression regimens used it is difficult to evaluate the independent effect of each regimen. However, given the known relation between treatment and the control of glomerular disease activity, it is likely that the accelerated renal function decline may be related to COVID-19, especially given the graded impact correlating with infection severity. Acknowledging these limitations, we have updated the limitation section of the study accordingly. 

2.It appears that SARS-CoV-2 infection may have induced a form of AKI and even many sentences that specify the effect of SARS-COV-2 and kidney that implies to this study should have more proof and state in the manuscript.

Response: Thank you for the observation. Indeed, the relation between SARS-CoV-2 infection and AKI in our study was just a hypothesis given the observation that the maximal impact on GFR loss in this study was seen at 12 months (and in other studies at 6 months). However, we could not definitely prove this aspect due to the absence of AKI assessment episodes and lack of evaluation for potential AKI biomarkers. In agreement with this aspect, we have updated the discussion section accordingly. 

Reviewer 2

1. Was COVID-19 nucleic acid screening regularly performed during follow-up time (For example, every 2 weeks or at least 1 month) in the negative group. And how to determine whether the patients in the negative group were never infected with COVID-19 during the follow-up time?

Response: SARS-COV-2 infection screening was undertaken regularly in our center in this category of patients with a periodicity of 1-3 months. Our center’s protocol was to undertake a proactive screening for SARS-CoV-2 infection in this category of immunosuppressed patients via outpatient visits, hospital admissions, periodic telephone interviews and a thorough review of the electronic clinical health records during the study follow-up. Similar to previous studies, COVID-19 was defined as participants who reported “definitely” or “probably or suspected” COVID-19 or had a positive antigen or serological test for COVID-19. Patients that never had any suspicion for SARS-CoV-2 infection, had persistent negative testing and negative serologic work-up were considered SARS-CoV-2 negative. We have updated the methods section to describe this aspect. 

2. Please describe the number and cause of death of patients in the two groups respectively.

Response: 19 patients died during the study follow-up: 3 patients due to SARS-CoV-2 infection and 16 patients due to reasons unrelated to COVID-19 (12 patients due to cardiovascular causes, 3 patients due to underlying malignancy and 1 patient due to a non-SARS-CoV-2 infection). We have updated this in the result section.

3. Please describe the proportion of patients in the positive group who discontinued immunosuppressive therapy.

Response: Our center’s protocol was not to modify the immunosuppressive regimens during the COVID-19 pandemic. However, during a SARS-CoV-2 infection a temporarily reduction in the doses of immunosuppressive regimens (for up to 10-14 days following the resolution of the infection) was undertaken. We have updated this aspect in the methods section. 

4. Please describe the number of COVID-19 infections for each patient in the positive group.

Response: Nine patients were identified to have two SARS-CoV-2 infection during the study follow-up, which all the subsequent infection being mild in severity. We have added this information to the result section. 

5. Were there significant differences in 24-hour urinary protein between the two groups during the follow-up time?

Response: Thank you for the observation. Indeed, proteinuria is one of the most significant risk factors for CKD progression irrespective of the underlying etiology. However, because of the retrospective nature of this study and the initial mobility restriction in the context of the pandemic we had many missing proteinuria assessments. We acknowledge that this is a limitation of the study and updated this section accordingly. In addition, in a recent CureGN analysis (published during the review of this paper), the impact of SARS-CoV-2 of future GFR loss remained meaningful even after adjustment for baseline proteinuria. So, with this drawback, we still believe that the impact of SARS-CoV-2 on renal outcome would have remained significant even after proteinuria inclusion in the model. Moreover, given the suggestion to limit the number of variables in the Cox model, we would still have a limited number of variables to potentially adjust for. In addition, the relevance of proteinuria on renal outcomes may differ significantly between different glomerular disorders (e.g.: IgAN vs. MCD vs. MN).

6. The number of endpoint events in COX regression should be at least 10 times greater than the number of independent variables. In this paper, ESRD only occurred more than 30 times, so the independent variables of COX regression should be limited only to 3 to 4 terms.

Response: We appreciate your suggestions. In this cohort, we had 53 cases of ESRD and 76 cases of the composite endpoint (ESRD and more than 30% decrease in eGFR). Accordingly, in order to select the best prediction models, we modified our multivariate Cox regression models and chose a stepwise backward elimination method. According to this model, we were left with 6 variables in the analysis regarding the composite endpoint and 4 variables in the analysis regarding only ESRD. We have modified throughout the manuscript this new analysis and updated the methods section to highlight the method used for multivariate Cox regression.

---

## [Decision Letter · Decision Letter 1]

20 Nov 2023

PONE-D-23-25206R1The impact of SARS-CoV-2 infection on renal function in patients with biopsy-proven kidney diseasesPLOS ONE

Dear Dr. Obrisca,

Thank you for submitting your manuscript to PLOS ONE. After careful consideration, we feel that it has merit but does not fully meet PLOS ONE’s publication criteria as it currently stands. Therefore, we invite you to submit a revised version of the manuscript that addresses the points raised during the review process.

**The manuscript focuses on a topic of potential interest. While the study has been significantly improved, there are still some minor concerns that should be addressed. To mention some of them: i) explain better and report in the table the severity of disease; ii) indicate that the heterogeneity of the underlying pathology could make it difficult to evaluate the progression of renal disease; iii) the lack of urinalysis data, not only proteinuria, but also haematuria and/or leukocyturia that could limit the evaluation of the possible renal involvement in COVID-19 infection; iv) include the suggested references; v) define the acronyms in full at first mention and correct typos.**

We look forward to receiving your revised manuscript.

Kind regards,

Giuseppe Remuzzi

Academic Editor

PLOS ONE

Journal Requirements:

Reviewers' comments:

Reviewer's Responses to Questions

**Comments to the Author**

1. If the authors have adequately addressed your comments raised in a previous round of review and you feel that this manuscript is now acceptable for publication, you may indicate that here to bypass the “Comments to the Author” section, enter your conflict of interest statement in the “Confidential to Editor” section, and submit your "Accept" recommendation.

Reviewer #1: (No Response)

Reviewer #3: All comments have been addressed

2. Is the manuscript technically sound, and do the data support the conclusions?

Reviewer #1: Yes

Reviewer #3: Yes

3. Has the statistical analysis been performed appropriately and rigorously? 

Reviewer #1: Yes

Reviewer #3: Yes

4. Have the authors made all data underlying the findings in their manuscript fully available?

Reviewer #1: Yes

Reviewer #3: Yes

5. Is the manuscript presented in an intelligible fashion and written in standard English?

Reviewer #1: Yes

Reviewer #3: Yes

6. Review Comments to the Author

Reviewer #1: The author has answered all my questions .The answers seems to explain all the points to a certain extent.

Reviewer #3: Clinically this study may be interesting even if it presents several limitations already inserted by the authors, as requested by the reviewers, but which may make the correct evaluation of the end points difficult.

In addition to the retrospective aspect, widely underlined by the reviewers and authors in the manuscript, the heterogeneity of the underlying pathology could also make it difficult to evaluate the progression of renal disease.

Furthermore, the lack of urinalysis data, not only proteinuria, but also hematuria and/or leukocyturia that could help to evaluate the possible renal involvement in COVID 19 infection.

…….This study, the first to our knowledge to address the long-term impact of SARS-CoV-2 infection on renal function:……

-Mohammad Atiquzzaman, Jordyn R Thompson, Selena Shao, Ognjenka Djurdjev, Micheli Bevilacqua, Michelle M Y Wong, Adeera Levin, Peter C Birks. Long-term effect of COVID-19 infection on kidney function among COVID-19 patients followed in post-COVID recovery clinic in British Columbia, Canada. Nephrol Dial Transplant . 2023 Jun 22:gfad121. doi: 10.1093/ndt/gfad121. Online ahead of print.

The acronyms should be written in full first, there are some typos

In conclusion, we have identified a significant impact of SARS-CoV-2 infection on longterm renal function in patients with biopsy-proven kidney diseases that leadsto a greater decline of eGFR and a worse renal survival, with a graded impact correlating with infection severity…………

The authors should explain better and report in the table the severity of disease

Should be interesting to insert the data about serum electrolytes and acid-base balance over time to evaluate a possible association between COVID infection and CKD progression

Should be interesting to insert the inflammatory indexes which are generally associated with Covid 19 infection but can also be increased in CKD, and can persist even in long COVID.

-The PHOSP-COVID Collaborative Group. Clinical characteristics with inflammation profiling of long COVID and association with 1-year recovery following hospitalisation in the UK: a prospective observational study Lancet Respir Med. 2022 Aug; 10(8): 761–775.

7. PLOS authors have the option to publish the peer review history of their article (what does this mean?). If published, this will include your full peer review and any attached files.

Reviewer #1: No

Reviewer #3: No

---

## [Author Response · Author response to Decision Letter 1]

28 Nov 2023

Dear Plos One Editorial team, 

On behalf of the co-authors, I want to thank you for the opportunity of incorporating editorial and reviewer comments made in relation to our revised manuscript entitled “The impact of SARS-CoV-2 infection on renal function in patients with biopsy-proven kidney diseases”. We hope to have addressed all the comments and suggestions and believe that it has made our report clearer and more meaningful for publication. 

Sincerely, 

Bogdan Obrisca

Editor

The manuscript focuses on a topic of potential interest. While the study has been significantly improved, there are still some minor concerns that should be addressed. To mention some of them: i) explain better and report in the table the severity of disease; ii) indicate that the heterogeneity of the underlying pathology could make it difficult to evaluate the progression of renal disease; iii) the lack of urinalysis data, not only proteinuria, but also haematuria and/or leukocyturia that could limit the evaluation of the possible renal involvement in COVID-19 infection; iv) include the suggested references; v) define the acronyms in full at first mention and correct typos.

Response: We appreciate the suggestions made by the editor and reviewers in relation to our manuscript. We hope that based on these suggestions, our revised manuscript is clearer and our results more meaningful. Please find below our answers to the issues raised.

i) The SARS-CoV-2 infection severity was assessed as defined by the COVID-19 Treatment Guidelines Panel of the National Institutes of Health, and detailed accordingly in the methods section. In our study we have distinctly evaluated the impact of infection severity on renal outcomes (mild-moderate versus severe COVID-19) and detailed it in Supplemental Table II. In addition, the Cox regression analysis was done to evaluate the impact of mild-moderate infection versus no infection and severe infection versus no infection. To be clearer in the conclusion of our study we have modified it as:

“In conclusion, we have identified a significant impact of SARS-CoV-2 infection on long-term renal function in patients with biopsy-proven kidney diseases that leads to a greater decline of eGFR and a worse renal survival, with a graded impact correlating with infection severity, those with severe forms of COVID-19 having the worst renal outcome.”

ii) We have updated the limitations section indicating that the heterogeneity of the underlying pathology could bring limitations to evaluation of the renal disease progression. 

iii) Thank you for the observation. We have included this aspect in the limitations of the study section. However, as previously mentioned by the reviewers, we could not have been able to include the urinalysis in the multivariate Cox regression due to the suggestion to limit our variables included in the model. Nonetheless, the stratification of the analysis according to the underlying disorders and use of immunosuppression provided the same results, and we feel that this may be a surrogate marker of the underlying activity of the disease.

iv) We have included the suggested references. 

v) We have revised the entire manuscript for typing errors and acronyms.

We hope that we have addressed all the issues raised. 

Reviewer 3

1) Clinically this study may be interesting even if it presents several limitations already inserted by the authors, as requested by the reviewers, but which may make the correct evaluation of the end points difficult.

In addition to the retrospective aspect, widely underlined by the reviewers and authors in the manuscript, the heterogeneity of the underlying pathology could also make it difficult to evaluate the progression of renal disease.

Response: We have updated the limitations section indicating that the heterogeneity of the underlying pathology could bring limitations to evaluation of the renal disease progression. However, apart from the CureGN cohort that included only patients with MCD/FSGS/MN/IgAN, the rest of the population-based studies have evaluated patients with CKD based solely on eGFR and/or albuminuria, while a heterogeneity of an underlying pathology might be also present in these cases (but not taken into account due to the lack of etiological assessment). Despite being clearly a limitation of the study, we believe that our attempt provides a better characterization of the study population compared to previous studies. 

2) Furthermore, the lack of urinalysis data, not only proteinuria, but also hematuria and/or leukocyturia that could help to evaluate the possible renal involvement in COVID 19 infection.

Response: Thank you for the observation. We have included this aspect in the limitations of the study section. However, as previously mentioned by the reviewers, we could not have been able to include the urinalysis in the multivariate Cox regression due to the suggestion to limit our variables included in the model. Nonetheless, the stratification of the analysis according to the underlying disorders and use of immunosuppression provided the same results, and we feel that this may be a surrogate marker of the underlying activity of the disease. 

3) …….This study, the first to our knowledge to address the long-term impact of SARS-CoV-2 infection on renal function:……

-Mohammad Atiquzzaman, Jordyn R Thompson, Selena Shao, Ognjenka Djurdjev, Micheli Bevilacqua, Michelle M Y Wong, Adeera Levin, Peter C Birks. Long-term effect of COVID-19 infection on kidney function among COVID-19 patients followed in post-COVID recovery clinic in British Columbia, Canada. Nephrol Dial Transplant . 2023 Jun 22:gfad121. doi: 10.1093/ndt/gfad121. Online ahead of print.

Response: Thank you for the suggested reference that we included in the discussion section. However, this cohort had a mean eGFR of 88 ml/min, while less than 25% of the study cohort had an eGFR below 60 ml/min, much lower than the mean 52 ml/min in our study, suggesting that we included a population with a higher risk for CKD progression. Nonetheless, similar to our study, this reference highlights the risk of eGFR loss within the first year following COVID-19 and this strengthens our similar observations. In addition, apart from the CureGN cohort that included a limited number of underlying CKD pathology, our study remains the first one to include a well characterized patient population with regards to underlying pathology, that rather reflects a population typical for a tertiary clinic. 

4) The acronyms should be written in full first, there are some typos.

Response: We have revised the entire manuscript for typing errors and acronyms. 

5) In conclusion, we have identified a significant impact of SARS-CoV-2 infection on long term renal function in patients with biopsy-proven kidney diseases that leads to a greater decline of eGFR and a worse renal survival, with a graded impact correlating with infection severity…………

The authors should explain better and report in the table the severity of disease.

Response: The SARS-CoV-2 infection severity was assessed as defined by the COVID-19 Treatment Guidelines Panel of the National Institutes of Health, and detailed accordingly in the methods section. In our study we have distinctly evaluated the impact of infection severity on renal outcomes (mild-moderate versus severe COVID-19) and detailed it in Supplemental Table II. In addition, the Cox regression analysis was done to evaluate the impact of mild-moderate infection versus no infection and severe infection versus no infection. To be clearer in the conclusion of our study we have modified it as:

“In conclusion, we have identified a significant impact of SARS-CoV-2 infection on long-term renal function in patients with biopsy-proven kidney diseases that leads to a greater decline of eGFR and a worse renal survival, with a graded impact correlating with infection severity, those with severe forms of COVID-19 having the worst renal outcome”

We hope that this addressed your issue.

6) Should be interesting to insert the data about serum electrolytes and acid-base balance over time to evaluate a possible association between COVID infection and CKD progression.

Response: Although we do agree that evolution of electrolytes and acid-base balance would have been very interesting to assess, the retrospective nature of the study and the fact that some patients were followed remotely during the lock-down period makes very difficult to collect this huge amount of data. In addition, many patients were already excluded from the study due to a lack of an adequate assessment of renal function due to various reasons. Including electrolyte/acid-base assessment would have further limited the number of patients that would have been eligible for study inclusion. However, we agree that this aspect would be very interesting and ideal to be assessed in a prospective cohort. 

7) Should be interesting to insert the inflammatory indexes which are generally associated with Covid 19 infection but can also be increased in CKD, and can persist even in long COVID.

-The PHOSP-COVID Collaborative Group. Clinical characteristics with inflammation profiling of long COVID and association with 1-year recovery following hospitalisation in the UK: a prospective observational study Lancet Respir Med. 2022 Aug; 10(8): 761–775.

Response: Thank you for the suggestion and we included this reference in the discussion section. However, similar to the previous response, the retrospective nature of the study and the fact that some patients were followed remotely during the lock-down period makes it very difficult to collect this nature of data.

---

## [Decision Letter · Decision Letter 2]

7 Dec 2023

The impact of SARS-CoV-2 infection on renal function in patients with biopsy-proven kidney diseases

PONE-D-23-25206R2

Dear Dr. Obrisca,

We’re pleased to inform you that your manuscript has been judged scientifically suitable for publication and will be formally accepted for publication once it meets all outstanding technical requirements.

Kind regards,

Giuseppe Remuzzi

Academic Editor

PLOS ONE

Additional Editor Comments (optional):

**Both the reviewers are pleased with the implemented changes.**

Reviewers' comments:

Reviewer's Responses to Questions

**Comments to the Author**

1. If the authors have adequately addressed your comments raised in a previous round of review and you feel that this manuscript is now acceptable for publication, you may indicate that here to bypass the “Comments to the Author” section, enter your conflict of interest statement in the “Confidential to Editor” section, and submit your "Accept" recommendation.

Reviewer #1: All comments have been addressed

Reviewer #3: All comments have been addressed

2. Is the manuscript technically sound, and do the data support the conclusions?

Reviewer #1: Yes

Reviewer #3: Yes

3. Has the statistical analysis been performed appropriately and rigorously? 

Reviewer #1: Yes

Reviewer #3: Yes

4. Have the authors made all data underlying the findings in their manuscript fully available?

Reviewer #1: Yes

Reviewer #3: Yes

5. Is the manuscript presented in an intelligible fashion and written in standard English?

Reviewer #1: Yes

Reviewer #3: Yes

6. Review Comments to the Author

Reviewer #1: (No Response)

Reviewer #3: The requested changes have been made , therefore the manuscript can be accepted in its current form

7. PLOS authors have the option to publish the peer review history of their article (what does this mean?). If published, this will include your full peer review and any attached files.

Reviewer #1: No

Reviewer #3: No

---

## [Editor Report · Acceptance letter]

13 Dec 2023

PONE-D-23-25206R2 

PLOS ONE

Dear Dr. Obrisca, 

I'm pleased to inform you that your manuscript has been deemed suitable for publication in PLOS ONE. Congratulations! Your manuscript is now being handed over to our production team.

Kind regards, 

on behalf of

Prof. Giuseppe Remuzzi 

Academic Editor

PLOS ONE